# Novel Polydimethylsiloxane (PDMS) Pulsatile Vascular Tissue Phantoms for the In-Vitro Investigation of Light Tissue Interaction in Photoplethysmography

**DOI:** 10.3390/s20154246

**Published:** 2020-07-30

**Authors:** Michelle Nomoni, James M. May, Panayiotis A. Kyriacou

**Affiliations:** Research Centre for Biomedical Engineering, City, University of London, London EC1V 0HB, UK; james.may.1@city.ac.uk (J.M.M.); p.kyriacou@city.ac.uk (P.A.K.)

**Keywords:** photoplethysmography, phantoms, light, tissue, vessels, PDMS

## Abstract

Currently there exists little knowledge or work in phantoms for the in-vitro evaluation of photoplethysmography (PPG), and its’ relationship with vascular mechanics. Such phantoms are needed to provide robust, basic scientific knowledge, which will underpin the current efforts in developing new PPG technologies for measuring or estimating blood pressure, blood flow and arterial stiffness, to name but a few. This work describes the design, fabrication and evaluation of finger tissue-simulating pulsatile phantoms with integrated custom vessels. A novel technique has been developed to produce custom polydimethylsiloxane (PDMS) vessels by a continuous dip-coating process. This process can accommodate the production of different sized vessel diameters (1400–2500 µm) and wall thicknesses (56–80 µm). These vessels were embedded into a mould with a solution of PDMS and India ink surrounding them. A pulsatile pump experimental rig was set up to test the phantoms, where flow rate (1–12 L·min^−1^), heart rate (40–120 bpm), and total resistance (0–100% resistance clamps) could be controlled on demand. The resulting flow profiles approximates human blood flow, and the detected contact PPG signal (red and infrared) from the phantom closely resembles the morphology of in-vivo PPG waveforms with signal-to-noise ratios of 38.16 and 40.59 dB, for the red and infrared wavelengths, respectively. The progress made by this phantom development will help in obtaining new knowledge in the behaviour of PPG’s under differing flow conditions, optical tissue properties and differing vessel stiffness.

## 1. Introduction

Photoplethysmography (PPG) [1], an optical technique first established in 1937 [2], is used to measure various subcutaneous haemodynamic properties with well-established clinical monitoring applications [3]. In principle, PPG’s are obtained from a simple sensor that transmits and detects light from a volume of tissue. The arteries and arterioles within tissue contain more blood during systole than during diastole. This change of blood volume causes absorption variation of photons and results in the PPG waveform [4]. The introduction of the pulse oximeter in 1974 advanced the use of PPG dependant technology significantly as it was adopted into anaesthetic practice [5]. Pulse oximeters take advantage of arterial pulsations to discriminate between arterial blood and other tissue absorbers and they determine the arterial oxygen saturation (SpO2) from the ratio of the absorption of photons at two wavelengths (red and infrared). Since its’ invention, pulse oximeters have become standard equipment in clinical environments [6].

The resurgence of PPG technology in the last few decades has been due to the growing demand for personal wearable health technologies, for example smart watches or sport-bands that enable the non-invasive assessment of cardiovascular physiology and potential disease diagnosis [7]. Despite the long-established history of PPG, research into the many causal factors remains a popular area of research [4,8,9]. The light-tissue interaction involved is a complex process of absorption, scattering, reflection and transmittance and thus requires in-depth investigation to understand the PPG’s true origin [10,11]. Detailed knowledge of the light tissue interaction will further push the scope of PPG sensors, more than just common heart rate and SpO2 measurement devices.

A common approach to understanding light-tissue interaction is through the application of computer simulations. Chatterjee et al. [12] utilise Monte Carlo simulations to examine the distribution of photons within perfused skin. They concluded that the Differential Pathlength Factor (DPF) is dependent on wavelength and source-detector separations contrary to prior assumptions [13]. Although such studies are producing interesting results, a common limitation is the static nature of the simulation design where arterial pulsations are ignored. In-vitro investigations address this limitation as pulsatile flow is easily simulated using pulsatile flow rigs. In-vitro investigations also have many advantages over in-vivo studies, where control over physiological parameters are impractical. Many groups are currently researching the potential use of PPG for non-invasive continuous measurement of blood pressure [14,15]. Varying patient’s blood pressures is a major limiting factor in in-vivo studies (especially in studies using healthy volunteers), so the employment of an advanced in-vitro rig where physiological parameters such as blood pressure can be precisely controlled could be an advantage.

There is large interest in developing a system that accurately characterises the true biology of tissue in a more controlled and rigorous fashion. A method to achieve this is using advanced phantom technology. The development of diagnostic systems for applications in healthcare, such as monitoring blood oxygenation, glucose concentration, in-vivo lactose measurement and several other biological markers through absorption spectra analysis, have all required the use of tissue-simulating phantoms [16,17,18].

Skin and vessel phantoms have been widely used as test models for a variety of peripheral tissue imaging techniques. These often comprise of a skin mimicking material surrounding a vessel through which blood mimicking fluid can be pumped or perfused. These phantoms can serve many purposes, a main feature would be to use it as a constant reference to compare existing systems against each other. They could also serve functions such as initial systems testing and optimisation of signal-to-noise ratios (SNR) during their development stage. The literature on materials used in phantoms is broad [19] and throughout the years several custom multi-modal phantoms have been described. Chen et al. [20] describes in-depth the effects of India ink, Intralipid, glycerol, gelatine and other substances as absorption and scattering attenuators primarily used in phantoms. Materials often used such as silicone offer long term stability in phantoms, providing consistency and accuracy of measurements over a prolonged period.

Regulatory bodies such as the American College of Radiology (ACR) often approve new systems based on their performance against quality control phantoms. This is an established criterion in medical imaging devices such as MRI, Ultrasound and CT machines. These phantoms imitate the selective properties of human tissues to test individual diagnostic modalities such as in ultrasound phantoms that are designed to imitate the geometric and acoustic properties. To design a phantom for PPG research the optical, mechanical and geometric properties all must be representative of the desired tissue to be investigated. There are limited optical phantoms available commercially that could be used in PPG based technologies to allow in-depth investigation of light tissue interaction. Commercial manufacturers with their considerable resources and production experience can produce high quality phantoms for industrial purposes in most modalities except for optical sensor technologies. Most optical phantoms are produced during the developmental stage of new systems at research institutes, where they have been found to be inconsistent, with their properties changing over time, and have low reproducibility [21].

The increased research associated with PPG morphology requires the use of vessel and tissue phantoms with tune-able properties to investigate different physiological conditions [22,23]. PDMS phantoms provide high stability and repeatability, and by varying the design of the phantom’s layers, different physiology can be represented.

In this paper we focus on the design, fabrication and characterisation of a pulsatile tissue phantom that mimics the mechanical and optical properties of the finger, a common site for PPG measurements. Ultimately, the investigations in this paper demonstrate the ability of the in-vitro system to be utilised in potential investigations of PPG sensor technologies against a pulsatile phantom where properties of the phantom or the pump can be altered to investigate different pathologies.

## 2. Materials and Methods

This section details further the novel technique used to fabricate custom PDMS vessels with comparable wall thicknesses to real blood vessels described in the groups previous work [24]. The placement of these vessels into a medium of artificial tissue is described, and the pump system design that allows parameter control is detailed.

### 2.1. Fabrication of Custom Vessels

The phantom was modelled after the finger, where PPG sensors are typically placed for pulse oximetry [1]. Each human finger is supplied with blood from two digital arteries, the radial and ulnar digital arteries. Leslie et al. [25] measured the average diameter of the largest digital artery to be 1.44 mm. The intima-media thickness (IMT) of digital arteries is not published, as far as we can tell, but the nearest vessel, the superficial palmar arch artery is measured to have an IMT of 105 µm [26]. No commercial artificial vessels or tubing is available at these values, or they are prohibitively expensive to have custom-made. Many other research groups have also faced challenges using commercially available vessels and have endeavoured to fabricate their own custom vessels from varying materials and methods [20]. Therefore, a unique fabrication method has been developed by our research team.

To produce a thin walled vessel, a continuous dip coating technique has been developed. A precision dip-coating machine (Qualtech Products Industry, Manchester, UK) was adapted to fabricate the vessels. Silicone tubing of 1.3 mm ID (Hilltop-Products, Warrington, UK), was used as a preform for the vessels. It was attached to the arm of the dip-coater and threaded through a heating coil. It was then passed through a trough. The tubing was hung across a series of pulleys with a tension weight, to help maintain an even coating by eliminating any geometrical irregularities caused by the tendency of the silicone form wanting to coil up under its own tension. The trough was filled with PDMS (Sylgard 184) (Dow Silicones, Barry, Wales, UK) chosen for its long pot life of two hours and fast cure rate. Figure 1 is a diagrammatic representation of this setup. A long pot life ensures high yield per batch before replacement of the PDMS is required, and the fast cure-rate results in a shorter time required passing through the heating coil allowing faster withdrawal rates. Once the system arrangement was complete, the heating-coil was switched on to increase the air temperature to 275 °C, verified with a handheld IR imaging device (FLIR Systems, Wilsonville, OR, USA). To determine the optimum withdrawal rate, the dip-coater system drew the tubing at steady rates between 5 and 20 mm·min^−1^, in 5 mm·min^−1^ intervals to evenly coat the tubing. The coated section of tubing then passed through the heating element, where the coating cured. After the dip coater reached maximum withdrawal length, the tubing was cut, and the arm was reset back to its minimum position. The cut length of coated tubing was placed in an infrared reflow oven set at 150 °C for 10 min to ensure a complete and thorough cure. The coating was then separated from the internal tubing. This method produced custom vessels with inner diameters equal to the outer diameter of the silicone form, and with wall thicknesses that can be adjusted by varying the speed of withdrawal. Figure 2 compares the commercial vessel used as the internal tubing (a) and the custom tubing produced (b) through the method described. 

### 2.2. Phantom Tissue

PPG measurements are often taken from the finger, and this site provides a convenience that can be utilised by personal health monitors. The finger consists of digital arteries that sit within fat and tendons with varied optical and mechanical properties. To simplify the phantom design at this stage, the chosen model consists of the custom vessels described in this paper surrounded by homogeneous tissue that closely resembles the average optical and mechanical profiles of human tissue. Lui et al. [27] produced a formula that can be used to calculate an accurate amount of India ink to be used, that would produce the desired absorption coefficients for the red and infrared wavelengths being tested, and here we use that formula to calculate the corresponding concentrations of additives to produce the absorption coefficients of subcutaneous fat [28]. Scattering properties were ignored at this stage to simplify the manufacture. The PDMS (Sylgard 184, Dowsil, UK) has a recommended catalyst ratio of 10%. At this ratio, the resultant hardness of the silicone is 43 shore A, much harder than human tissue. To determine the correct catalyst ratio, the shore harness (scale 00) was measured at all ten fingertips from ten volunteers and averaged. The results are shown in Table 1. The catalyst was empirically determined to require a 3% ratio to yield a shore hardness value close to that of the tissue at fingertips.

### 2.3. Phantom Assembly

A rectangular container with the approximate dimensions of a finger was 3D printed (Formlabs, Somerville, MA, USA). The 70 × 15 × 15 mm open top container consisted of holes on either end to thread the vessel through. Three vessels (ID = 1.4 mm) were passed through the container on one side and were secured in place using UV hardened high temperature glue. The other end of the vessel was stretched taut and secured. The glue acted as a plug to avoid drainage during the curing process. The silicone tissue mixture was then poured into the container and cured at 150 °C for 10 min. The phantom was then cooled before attaching the vessels to connectors and the fluid distribution network. The phantom was secured onto an acrylic sheet to avoid accidental damage, as seen in Figure 3.

### 2.4. In-Vitro System Setup

The set up for the in-vitro experiment follows the Windkessel model, a widely used model for estimating the overall compliance of a systematic arterial system [29]. Figure 4 shows the actual physical setup of the pump with the diagrammatic representation of all major components. The pump is a pulsatile fluidic pump from BDC Labs (Wheat Ridge, CO, USA), driven by a computer-controlled linear motor. Isolation between the pump head and pulsatile fluid is achieved by a fluid isolation module. This facilitates the pumping of liquids that may otherwise interfere with the operation of the motor, including biological fluids (blood, serum, and plasma), fluids with particle suspensions and Phosphate Buffered Saline (PBS) solutions. The pulsatile fluid sits in a chamber that can be pressurised and heated to effectively control overall compliance and offer control over liquid viscosity. The outlet feeds an artificial Aortic vessel (ID = 40 mm, vulcanised rubber). A single tap off the Aortic vessel feeds a brachial vessel (ID = 2.4 mm, silicone). The brachial vessel is stepped down to the finger tissue-vessel phantom models under test. The outlet of the phantom is stepped back up to a brachial vessel before returning to the vena-cava vessel on the opposite side of a resistance clamp that is designed to simulate the reflection of the Iliac artery (Iliac Reflection Clamp IRC). Before returning to the pump for recirculation, the fluid passes a total peripheral resistance (TPR) clamp designed to simulate whole body peripheral resistance on the pulsatile fluid. Figure 4 shows the systems setup with inline pressure transducers placed on the compliance chamber, to the input of the brachial feeding the phantom, and on the outlet of the phantom.

## 3. Results

### 3.1. Custom Vessels

The custom vessel production was a new technique developed as a solution to limitations found in vascular tissue phantom development [20]. The main variable during production was the withdrawal rate. Four speeds were selected and tested, 5, 10, 15, and 20 mm·min^−1^. The vessels were sliced at varying points along their lengths and placed under a microscope (Micropix ltd, Midhurst, UK) to take measurements using YenCAM (Micropix ltd, Midhurst, UK), see Figure 2. Analysis was performed using MATLAB (The MathWorks, Plano, TX, USA) to determine the effect of withdrawal speed on wall thickness and variability. The boxplot in Figure 5 displays the distribution of wall thickness at the withdrawal speeds of 5 to 20 mm·min^−1^. The median wall thickness shows a positive correlation with withdrawal speed and demonstrates the effect that withdrawal speed has on the variability of wall thickness measurements. Both commercial (ID = 1.3 mm) and custom vessels (ID = 1.35 mm) were embedded into separate phantom moulds and encased within clear silicone tissue. Clear tissue was chosen at this stage to eliminate further variables affecting the resultant signals. The phantoms were connected to the in-vitro system and a pulsatile flow was introduced into the phantoms. PPG data was collected from the sensor (OSRAM SFH 7050) placed above the surface of the phantom. The PPGs of both phantoms were recorded in Labview (National Instruments, Austin, TX, USA) and are compared in Figure 6.

### 3.2. In-Vitro System

Using the Windkessel model [30] as a reference guide to the arrangement of the pump circuit, the pump actuator was driven with a bell curve and the pressure transducers were connected at three different locations, see Section 2.4. Figure 7 displays the driving signal of the pump with the consequent pressure signals in the fluid chamber, pre-phantom and post-phantom locations. The pump was set to output a flow of 7 L·min^−1^ at 60 bpm. The results demonstrate that the dicrotic notch visible at pre-phantom and post phantom are a direct result of the resistance elements, as the dicrotic notch is not witnessed inside the fluid chamber.

To qualitatively study the effectiveness of the resistance elements in our pump circuit, PPG signals were taken from the clear sample phantom with custom vessels at the four resistance configurations. The pump was set to output a flow of 7 L·min^−1^ at 60 bpm. Figure 8 shows the resulting PPG signals obtained from the phantom with the resistance elements open (unrestricted) and closed (restricted). The restricted position of each resistance element was an 80% reduction in the vessel width from 40 to 8 mm.

It is important to understand the relationship between heart rate, flow rate and the resulting blood pressures in the system. The resultant pressure amplitudes pre-phantom of different heart rates and flow rates are presented in Figure 9. Figure 9a shows the resulting blood pressures of heart rates ranging from 40 to 120 bpm at a flow rate of 6 L·min^−1^. Figure 9b shows the resulting blood pressure of flow rates ranging from 1 to 12 L·min^−1^ at 60 bpm. These graphs show the maximum and minimum recorded pressure amplitudes at each level, with the calculated mean arterial pressure (MAP) indicated.

### 3.3. PPG Analysis

The phantom described in Section 2.2 was used to take PPG results at the range of heart rate and flow rate values described in Figure 9. The Raw AC signals recorded from the ZenPPG acquisition system [31] were digitally filtered and the mean amplitude of three-minute recordings were obtained. Figure 10 displays the mean PPG amplitudes of windowed signals at varied heart rates. The trend line observed shows there is a negative correlation. Figure 11 shows the positive correlation between flow rate and mean PPG amplitude. 

## 4. Discussions

For the first time, a vessel-tissue phantom has been produced with geometrically and mechanically accurate vessels. When integrated into a pulsatile pump, the tissue exhibited similar morphological features to a physiological PPG signal with similar signal-to-noise ratios [32].

As a result of previous work [33] and the highlighted issues with commercially available tubing, a new method for vessel production was developed. The custom-made vessels were withdrawn at different speeds that helped identify two factors, the coating period and the curing period. Firstly, the coating period was determined by the time between the vessel leaving the solution and when it enters the heating element. This had a role on the final thickness of the coating which in turn would be the wall thickness of the final vessel. Secondly, the curing period was determined by the time the coated vessel spent curing in the heating element. If the vessel did not spend enough time in the heating element, it would not fully cure, and variability increased as a result. Experimentally, the 5 mm·min^−1^ speed produced vessels with the highest variability, and 20 mm·min^−1^ produced the thickest walls. The results show that a speed of 10 mm·min^−1^ was best to provide an even cure across the length of vessels and that faster speeds had little impact on wall thickness but did increase the variability of wall thickness on a single vessel.

Chen et al. [20] discussed in their phantom development process that commercial vessels were a major hindrance in phantom quality and that future work would involve self-fabrication of vessels. Through our novel continuous dip coating method, we have fabricated vessels with properties that resemble tissue more closely than commercially available silicone vessels. The cross section of the commercial and custom vessel shows the striking difference in wall thickness achieved. The commercial silicone vessels are measured to have a 350 µm wall thickness and the custom-made vessels withdrawn at a speed of 10 mm·min^−1^ have a median wall thickness of 60 µm. The process can be adapted to accommodate the production of different diameters and vessel wall thickness with simple changes. This flexibility is the major advantage of this vessel self-fabrication method.

The PPG signals obtained from the phantom with the commercially available tubing embedded had low SNRs, 5.38 and 10.59 dB in red and infrared wavelengths, respectively, with no morphology of the PPG waveform observed. In comparison, the custom-made phantoms described in this work produced high SNRs of 38.16 and 40.59 dB for the red and infrared wavelengths, respectively. The reduction in vessel wall thickness from the commercial to custom vessel improves the SNR significantly, demonstrating that the elasticity, and therefore compliance has also increased. The fabrication method of custom vessels described in this paper has produced vessels that approach the mechanical properties and physical dimensions of the digital arteries.

The in-vitro system was set up to allow individual control over its parameters. Through the adaptation of a Windkessel model, used to describe the function of the heart and the compliance of arteries to produce the arterial pressure wave, a versatile in-vitro pump circuit has been developed. A bell curve drives the actuator, pushing fluid from the chamber into the circuit. A pressure transducer placed in the fluid chamber shows the pressure waveform to be 15 mmHg in amplitude with a steep incline and sharp peak at 125 mmHg. The fluid then branches off into the phantom where the second pressure transducer measures an amplitude of 15 mmHg but has a max peak of 122 mmHg, a small reduction from the pressure in the chamber. The waveform exhibits a dicrotic notch, caused by the distended tubing recoiling. The pressure wave loses energy as it travels through the phantom and is measured to have an amplitude of 10 mmHg peaking at 97 mmHg at the junction where flow re-enters the main circuit.

The morphology of the acquired PPG was dependant on the configuration of the resistance elements. When both elements remain unrestricted, the PPG displays no notch. Restriction of the TPR element only, produced a notch in the waveform and resulted in a lower amplitude PPG signal. Restriction of the IRC element caused a large increase in the systolic peak of the PPG, where the notch is dramatically defined. The combined restriction of both elements produced a waveform that closely resembles the “textbook” PPG/blood pressure waveform.

In the in-vitro setup, the pump software allowed the user to input the heart rate and flow rate. These parameters and their effect on pressure were investigated. Increasing the heart rate had little impact on the mean arterial pressure, it largely affected the systolic and diastolic values. Increasing the heart rate increased the diastolic pressure but decreased the systolic pressure. So, although MAP remains steady, the peak to peak amplitude of the pressure signal was reduced. The biological response is comparable to this result, as the human heart rate fluctuates throughout the day, our blood pressure remains mostly unaffected by it. The compliance of arteries can adapt to the changing heart rate and regulate blood pressure. The investigation proves that the in-vitro system can regulate a steady MAP at different heart rates, and the flow rate impacts the pressure the most.

PPG’s obtained at all the heart rate and flow rate values implemented were plotted and the trends displayed fitted the expected outcome of the volumetric model. Chan et al. [34] describe the fundamental principle of PPG where the AC component of the signal is mainly attributed to the volume changes that occur in the vessels. Increasing the heart rate, ultimately reduced the amplitude between systole and diastole (see Figure 10), which in turn reduced the distention of the vessels. This resulted in less volume of fluid therefore reducing the amplitude of the PPG. The volumetric model is again evident in the flow rate investigation. The increase of flow rate resulted in a larger volume of fluid per stroke. This positive correlation is seen in Figure 11.

## 5. Conclusions

In this paper, we have discussed the design and fabrication of a pulsatile tissue phantom, employing a novel technique to produce custom silicone vessels. Our results showed for the first time an optical tissue phantom in which its properties mimic the anatomy of tissue (in this case the finger) regularly chosen for PPG measurements. We have demonstrated the versatility of our setup and how physiological parameters such as heart rate, flow rate and clamp positions impact the resultant pressure and PPG signals. The development of this phantom, and its validation through fundamental investigations are a step towards the development of a phantom where the precise nature of light tissue interactions can be examined under different mechanical and optical properties and the theories on the exact origin of the PPG can be further investigated and tested.

Lastly, there is no current criterion or industry standard phantom for which PPG technology is assessed against. Adopting common performance evaluation protocols for optically based medical instruments will help deploy new technologies for clinical use more rapidly by improving their reliability and results reproducibility. Such advanced tissue phantoms can also play a pivotal role in the calibration of pulse oximeters, as such devices are primarily calibrated using in-vivo blood sampling techniques during progressive hypoxia protocols in healthy volunteers.

## Figures and Tables

**Figure 1 sensors-20-04246-f001:**
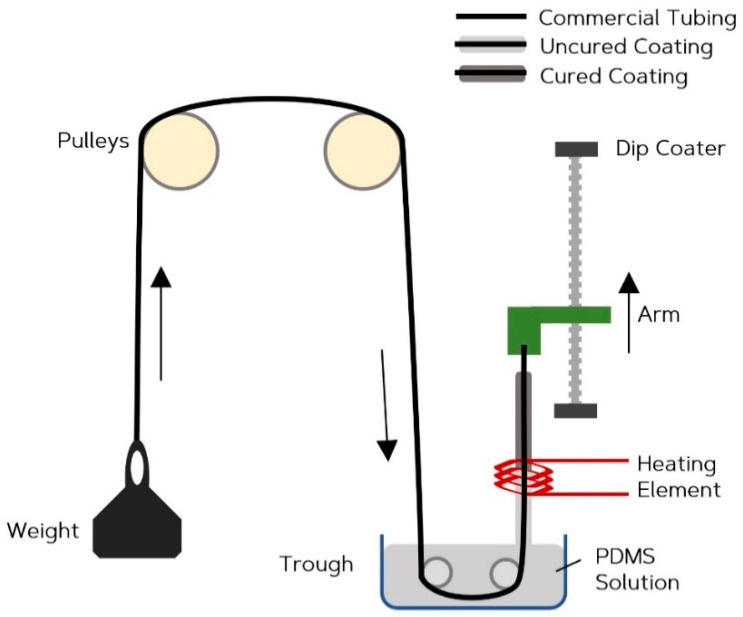
Continuous dip coating fabrication method.

**Figure 2 sensors-20-04246-f002:**
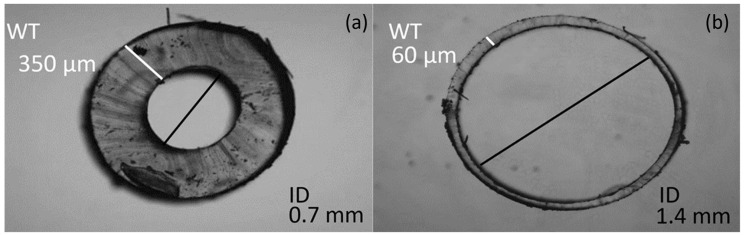
Images of vessel cross sections. Commercial vessel (ID = 0.7 mm) (**a**) Custom Vessel (ID = 1.4 mm) (**b**) withdrawn at 10 mm·min^−1^. (ID = inner diameter).

**Figure 3 sensors-20-04246-f003:**
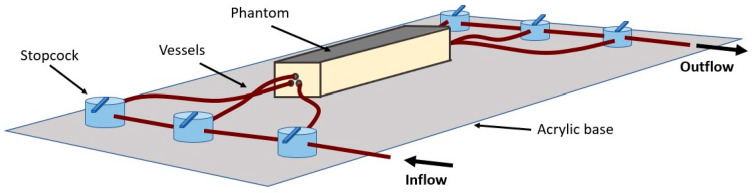
Illustration of the assembled phantom connected to the distribution network fixed on an acrylic sheet.

**Figure 4 sensors-20-04246-f004:**
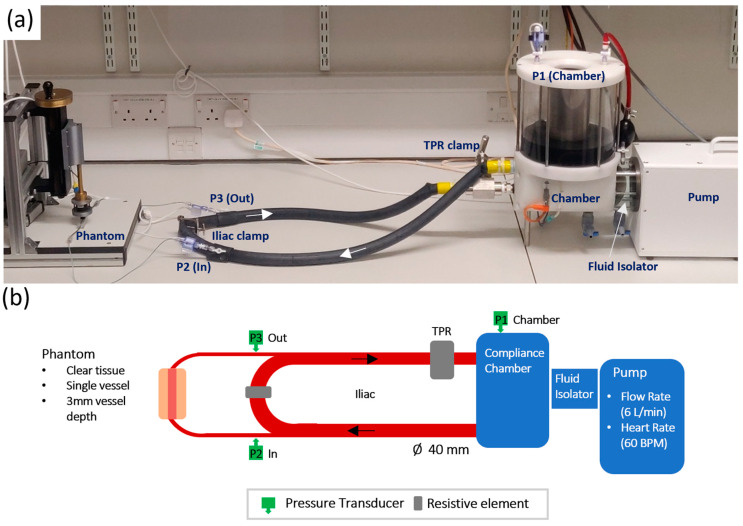
Pump setup. Actual in-vitro lab setup (**a**), with diagrammatic representation of all major components (**b**).

**Figure 5 sensors-20-04246-f005:**
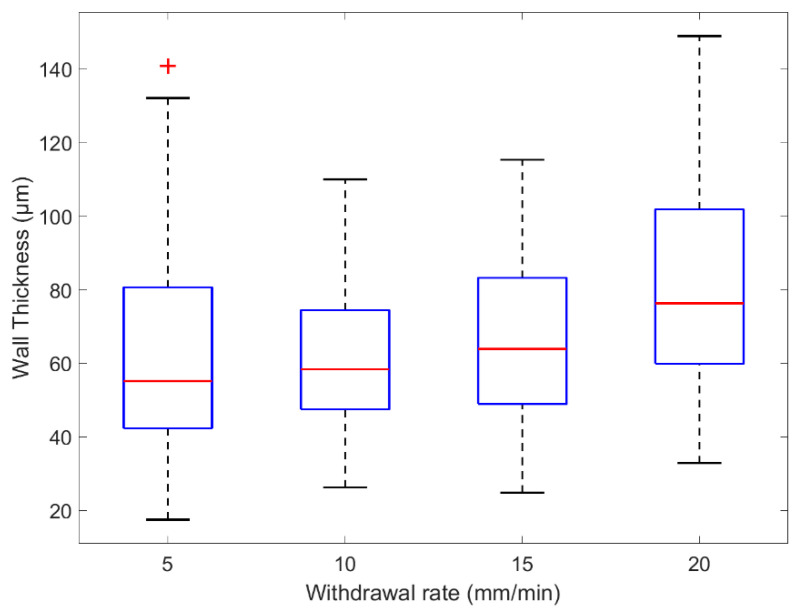
Box plot of wall thicknesses measured from custom vessels withdrawn at several rates.

**Figure 6 sensors-20-04246-f006:**
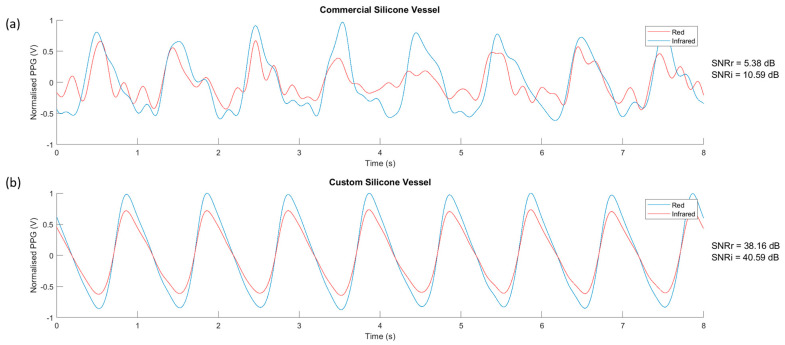
The photoplethysmography (PPG) signal from the clear vessel phantoms. (**a**) Commercial vessels, (**b**) Custom vessels. SNRr = signal-to-noise ratio (red wavelength). SNRi = signal-to-noise ratio (infrared wavelength).

**Figure 7 sensors-20-04246-f007:**
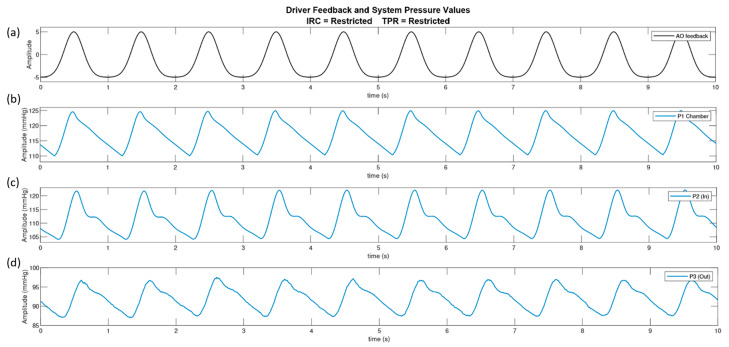
The analogue feedback from the driving waveform (**a**), from the linear driver and the pressure waves in the system at three locations; the compliance chamber (P1) (**b**), pre-phantom (P2) (**c**) and post-phantom (P3) (**d**).

**Figure 8 sensors-20-04246-f008:**
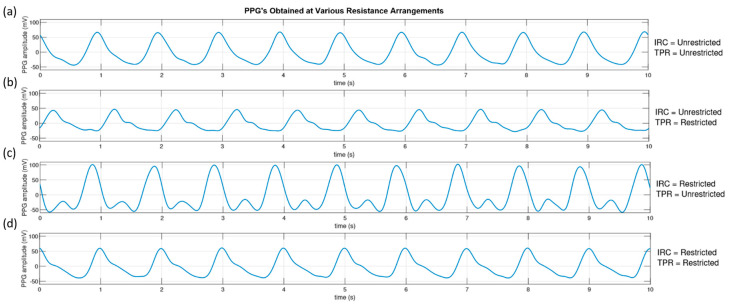
The pressure waves detected at the pre-phantom (P2) pressure transducer for different combinations of resistive elements positions. (**a**) Iliac Resistive Component (IRC) and Total Peripheral Resistance (TPR) both unrestricted. (**b**) IRC unrestricted, TPR restricted. (**c**) IRC restricted, TPR unrestricted. (**d**) IRC and TPR restricted.

**Figure 9 sensors-20-04246-f009:**
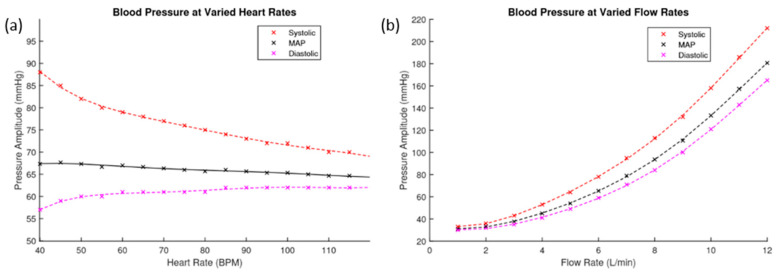
(**a**) This figure plots the blood pressure values pre-phantom against varied heart rates. (**b**) This figure plots the blood pressure values pre-phantom against varied flow rates.

**Figure 10 sensors-20-04246-f010:**
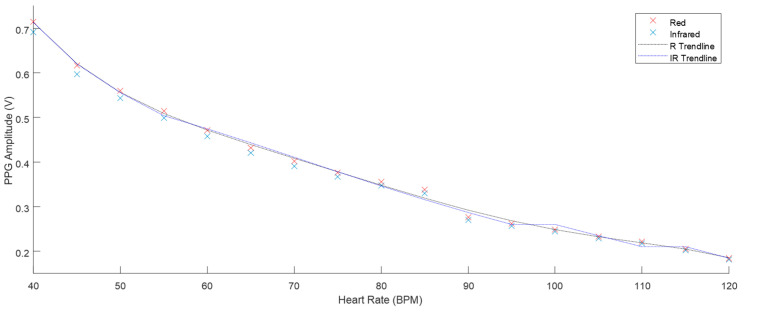
The mean PPG amplitude of three-minute samples with 6 L·min^−1^ flow and heart rates from 40–120 BPM (all SD < 0.003).

**Figure 11 sensors-20-04246-f011:**
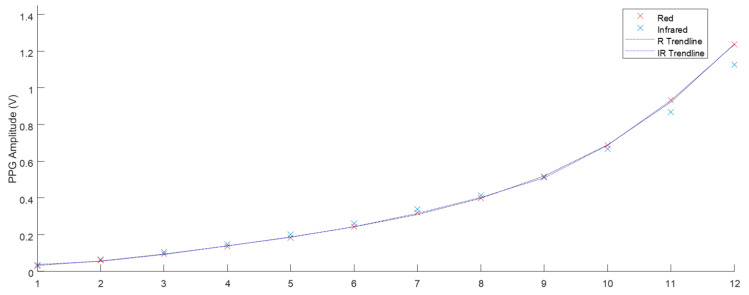
The mean PPG amplitude of three-minute samples with 60 BPM heart rate and flow from 1–12 L·min^−1^ (SD < 0.003).

**Table 1 sensors-20-04246-t001:** Results comparing shore hardness at each fingertip and the value for the polydimethylsiloxane (PDMS) when mixed with 3% catalyst. A catalyst ratio below 3% was found to be unstable. Measurements taken with a shore 00 durometer.

	Location	Mean	SD
Left	Thumb	24.4	3.3
	Index	28.7	4.1
	Middle	25.9	3.8
	Ring	23.9	3.5
	Little	29.3	3.4
Right	Thumb	25.4	3.7
	Index	29.4	3.7
	Middle	25.9	4.9
	Ring	23.4	3.0
	Little	31.8	4.2
Phantom	-	32.5	2.4

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
