# Peer review of "Novel Polydimethylsiloxane (PDMS) Pulsatile Vascular Tissue Phantoms for the In-Vitro Investigation of Light Tissue Interaction in Photoplethysmography"

_sensors, 2020, doi:10.3390/s20154246_

Round 1

Reviewer 1 Report

I thank the editors for the opportunity to review this article titled Novel polydimethylsiloxane (PDMS) pulsatile vascular tissue phantoms for the in vitro investigation of light tissue interaction in Photoplethysmography. In this article, the authors report a challenging and well-designed study into developing a device to simulate human blood vessels for the development and refinement of optical techniques used to measure the blood pressure, blood flow, arterial stiffness, and other physiological measures. The article reviews the need for a device that would allow investigators to finely (and broadly) control and modulate the blood pressure of the “patient”, which is impractical in humans, and as such requires some technological development.

The authors describe the manufacture of vessels made to mimic human arteries in the finger, with impressive precision, as confirmed my microscopic inspection of cross-sections of the “vessel”. The authors go on to describe the methods used to assemble the vessels into a model of the human finger. I am most impressed with the attention paid to full disclosure of the decision-making process and ensuring that the recipe is reproducible by the reader. This article represents a useful resource for the manufacture of a whole range of simulation devices using the materials available today.

The article concludes that their new method of manufacture affords a significant increase in SNR (by a factor of 4-6 dependent on sensor light wavelength). The entire experimental set-up was successful in simulating a finger that exhibited measurable blood flow with controllable heart rate and flow rate, and responses that convincingly mimic a biological system.

I have no major concerns about the validity of the science in the article. In fact, I reiterate that the methods are clear, thorough, transparent and replicable. Below are some comments that the authors may wish to integrate into future revisions of their work.

Somewhere around the paragraphs spanning lines 67-88 I would add a few sentences commenting on the use of phantoms for longitudinal QA and calibration on a system to ensure that accuracy and consistency of measurements are maintained over time.

Line 94 is the first mention of the word “finger”. I would strongly recommend getting this in the title and abstract, but at least the abstract.

Line 45 starts a new paragraph, and shouldn’t.

To be really picky, the sentence starting on line 51 is really clunky to read – I recommend rephrasing.

Please add to Figure 2 that ID = inner diameter.

Figure 5 – please change the x axis labels to reflect the speed. i.e. mm min-1 not just mm.

Line 300 – should it be “vessels” not “vessel”?

Author Response

Thank you for your review. I have made the changes you reccomened as follows:

1) I have added a short sentence regarding the stability of silicone phantoms. To add further insight into QA of longitudinal studies, it would require further reading into studies that have been carried out. This is something that will be included in a phantom review paper I am currently writing as it fits better there.

2) As advised I have inserted 'finger' into the abstract. line 15.

3) The paragraph break has been corrected.

4) The sentence on line 51 has been paraphrased to, 'They concluded that the Differential Pathlength Factor (DPF) is dependent of wavelength and source-detector separations contrary to prior assumptions'.

5) Figure 2 caption has been ammended to include ID= inner diameter.

6) Figure 5 has now been corrected with the correct axis label.

7) Indeed it should have been 'vessels'. its has now been corrected. line 305.

Reviewer 2 Report

The manuscript titled "Novel polydimethylsiloxane (PDMS) pulsatile vascular tissue phantoms for the in vitro investigation of light tissue interaction in photoplethysmography" describes the preparation of a vascularized tissue phantom with elastic blood vessels and a mimicry mock circulation system for in vitro PPG simulation. The authors successfully prepared a PDMS polymer blood vessel phantom using a dip coating system. Experimental rig with pulsatile pump simulating blood flow with various flow rates and heart rates showed interesting results for in vivo phantom-based PPG simulating system.
This manuscript is well prepared for the PPG phantom-based in vitro system establishment with a pulsatile blood analog.

There are a few points to be corrected or addressed prior to publication as below;
Page 2 line 51: abbreviation of DPF full expression needed
Figure 5: X-axis title needed. Statistical significance needed for the box charts. Recommend to use parentheses for unit expression after the Y-axis title (unit).
Figure 6: Designate full name for SNRi and SNRr, there is no explanation for the subscripts "i" and "r".
As the normalized graphs have to have the highest value 1, the graphs were not normalized or only the red graph of custom silicone vessel was normalized in Figure 6. Need to be adjusted either manuscript explanation or graphs.
Figure 7 Unit of X-axis title is needed.

Author Response

Thank you for your review. I have made the following changes based on your comments.

1) This sentence has been paraphrased and now expresses the abbreviation in full. DPF in line 51.

2) Figure 5 now has the correct units.

3) SNRi and SNRr have now been explained in the figure caption. Figure 6.

4) This was an error on my part. The data has now been correctly normalised. Figure 6.

5) The axis title and units are displayed under each graph. Time (s). Figure 7.

Reviewer 3 Report

The authors present a PPG study with newly developed tissue phantoms out of PDMS. The paper is well structured and the presented experimental results are convincing. However the editing of the manuscript shows some carelessness.

  1. There should be spacing inbetween the values and units, superscripts are missing,
  2. "et al." has to replace "et al"
  3. check for typos, e.g. line 59
  4. all abbreviations have to be introduced, e.g. DPF, line 51
  5. the mean values of Shore hardness are given with awkward precision (xx.x0)
  6.  in figure 5 the unit should be "µm" in the y-axis, for the x-axis it should be "mm/min"
  7. in the caption of figure 9 the graphs are denoted with "a" and "b", this should  be part of the figure
  8. the bibliographical information in some of the references is incomplete (e.g. 4, 10, 12,...)
  9. comma placement is not reasonable in the references

Author Response

Thank you for your review. I have made the following changes based on your comments.

1) I have corrected the units formatting throughout the paper. Superscripts included.

2) et al has now been corrected to et al.

3) Typo on line 59 has been corrected.

4) abbreviation has been added. DPF line 51.

5) The values in table 1 have all been adjusted to 1 d.p.

6) Figure 5 axis titles and units have been corrected.

7) The label of a and b have been added to figure 9.

8,9) The bibliography has been completed with the format changed to APA. It was completed using Mendeley.